# Evaluation of Growth, Physiological, and Biochemical Responses of Different *Medicago sativa* L. Varieties Under Drought Stress

**DOI:** 10.3390/plants14050639

**Published:** 2025-02-20

**Authors:** Yang Wang, Sisi Long, Jiyuan Zhang, Puchang Wang, Lili Zhao

**Affiliations:** 1College of Animal Science, Guizhou University, Guiyang 550025, China; wy1346791029@163.com (Y.W.); longsso3343@163.com (S.L.); zhangjiyuan010@163.com (J.Z.); 2School of Life Sciences, Guizhou Normal University, Guiyang 550001, China; wangpuchang@163.com

**Keywords:** alfalfa, drought stress, principal component analysis, membership function analysis, physiological and biochemical indicators

## Abstract

Alfalfa (*Medicago sativa*), an important leguminous forage crop, is valued for its high nutritional content, substantial yield, palatability, and broad adaptability. Drought is among the most significant environmental constraints on alfalfa growth, particularly in the karst regions of southwestern China. In this study, we conducted pot experiments to investigate the growth and physiological responses of seven alfalfa varieties introduced into the karst region of Guizhou under drought conditions. The results revealed that drought stress markedly reduced both plant height and aboveground biomass accumulation. Moreover, under drought stress, these alfalfa varieties exhibited increased root length, root surface area, and root tip number; elevated protective enzyme activities; and decreased levels of hydrogen peroxide (H_2_O_2_) and malondialdehyde (MDA), thereby maintaining relatively higher water content. Each of the seven varieties displayed distinct growth and physiological adaptation mechanisms under drought stress. Integrating principal component analysis and membership function analysis, we ranked the drought resistance of these alfalfa varieties from highest to lowest as follows: Crown > WL525 > Colosseo > Victoria > PANGO > Giant 801 > Dimitra. These findings provide valuable insights for introducing drought-resistant alfalfa varieties into karst regions of southwestern China and offer guidance for breeding and cultivation strategies across various environmental conditions.

## 1. Introduction

Water makes up approximately 80–95% of a plant’s fresh biomass and is vital for various physiological and biochemical processes [1,2]. In agricultural systems, plants frequently endure water deficiency, a stress recognized as one of the most critical environmental constraints on crop production [3,4]. This challenge is particularly pronounced in the karst regions of southwestern China, where the slow weathering of limestone severely limits soil water retention capacity. Despite relatively abundant rainfall, drought stress remains commonplace in these areas [5]. Such stress disrupts plant–water relationships, diminishes water-use efficiency, and impairs crop stem elongation, root proliferation, and leaf expansion, ultimately reducing yields and causing significant economic losses [6,7,8]. Therefore, selecting drought-tolerant crop varieties offers an effective strategy to bolster agricultural productivity and stabilize farmers’ incomes in these karst landscapes [9].

*Medicago sativa*, commonly known as alfalfa, is a perennial leguminous forage crop frequently referred to as the “king of forages” due to its high protein content for livestock [10]. Globally, alfalfa is cultivated on over 4 × 10^7^ hectares of land [11]. Within China, most alfalfa production occurs in the northwest, north, and northeast regions [10]. However, as the livestock industry continues to optimize and expand, alfalfa cultivation in southwest China—particularly in Guizhou Province—has gradually increased. Given Guizhou’s predominantly karst topography, drought conditions commonly affect alfalfa growth in this region [12].

As climate change intensifies the frequency and severity of droughts, understanding how alfalfa withstands water stress has become an essential area of research [13]. A growing body of literature indicates that drought stress significantly impacts alfalfa across multiple levels, including its morphology, physiology, and molecular biology [14]. Morphologically, drought stress results in reductions in root length, biomass, and leaf area, while also altering root shape and the number of root buds [15]. Physiologically, drought triggers the accumulation of reactive oxygen species (ROS), such as superoxide anion (O_2_^−^), hydrogen peroxide (H_2_O_2_), and hydroxyl radicals (OH^−^), which induce oxidative stress and impair metabolism and growth. To counteract this, alfalfa increases the activity of antioxidant enzymes, such as superoxide dismutase (SOD), peroxidase (POD), and catalase (CAT), which scavenge ROS and enhance drought tolerance [16,17]. At the molecular level, genes like glycosyltransferases (*UGTs*) and *MsTERT* play critical roles in modulating metabolic pathways that detoxify reactive intermediates and support the synthesis of secondary metabolites, contributing to stress resistance [18,19]. Previous studies have emphasized significant changes in alfalfa’s root morphology, microbial communities, and biomass under drought conditions compared to well-irrigated plants [20,21,22]. Existing studies have provided valuable insights into the physiological mechanisms of drought adaptation in alfalfa. However, despite these advances, research on drought-tolerant alfalfa varieties, particularly those suited to the karst regions of Guizhou, remains limited. This highlights the urgent need for targeted studies to address the unique regional challenges posed by drought in these areas.

The present study aims to address this gap by investigating the growth and physiological responses of seven alfalfa varieties that have been successfully introduced to Guizhou’s karst region. Specifically, the objectives are to: (1) quantify the varieties’ growth performance under drought stress, (2) determine the level of physiological indicators, and (3) identify drought-adapted varieties suitable for the karst landscapes of southwestern China. By integrating principal component analysis and membership function analysis [19], this study provides a robust theoretical framework for optimizing alfalfa cultivation in drought-prone karst environments. By focusing on drought-adapted varieties and their physiological responses, this work will help to select drought-tolerant alfalfa varieties in areas where climate change is leading to increasingly severe water deficiency.

## 2. Results

### 2.1. Effects of Drought Stress on Plant Height and Aboveground Biomass in Seven Alfalfa Varieties

Under normal watering conditions, plant height and aboveground biomass of seven alfalfa varieties increased over time. Under normal watering conditions, plant height and aboveground biomass differed significantly among varieties. Crown, Colosseo, and WL525 exhibited relatively high plant heights and aboveground biomass, whereas Dimitra, Giant 801, and PANGO displayed lower values (Table 1). In contrast, drought stress markedly inhibited increases in plant height and aboveground biomass across all seven alfalfa varieties (Table 1).

After 14 days of drought stress, plant height significantly decreased in five of the seven varieties (Crown, Colosseo, WL525, Victoria, and Dimitra) compared with the control (*p* < 0.05), whereas Giant 801 and PANGO showed no significant reduction (Table 1). Notably, Victoria experienced the largest decline, reaching only 57.8% of the control plant height. The alfalfa varieties Dimitra, WL525, and Crown exhibited reductions to 76.6%, 70.3%, and 60.9% of the control, respectively. By 21 days of drought stress, all plants of Dimitra had died. Compared with the control, Crown and PANGO showed the most pronounced decreases in plant height at this time, retaining only 57.4% and 60.9% of the control values, respectively, while Victoria and WL525 showed no significant difference in plant height compared to the control group.

After 14 days of drought stress, Victoria’s aboveground biomass significantly declined to 34.6% of the control. Dimitra’s aboveground biomass increased to 113.1% of the control, although this change was not statistically significant. Similarly, the other varieties did not exhibit significant changes. Following 21 days of drought stress, Victoria’s aboveground biomass was significantly lower than that of the control (31.4% of control biomass). These findings demonstrate that alfalfa varieties differ widely in their responses to drought stress; some varieties can better maintain plant height and aboveground biomass under adverse conditions.

### 2.2. Effects of Drought Stress on Root Morphology in Seven Alfalfa Varieties

#### 2.2.1. Root Length, Root Surface Area, and Root Volume

Root scanning images of the seven alfalfa varieties under normal watering and drought treatments for 21 days are shown in Figure 1. As illustrated in Figure 2, drought stress increased root length in six varieties compared to the control group, while Dimitra died. After 14 days of drought, Crown exhibited the longest root length. Compared with the control group, only Dimitra demonstrated a statistically significant increase in root length (*p* < 0.05), while the increase in root length of other varieties did not reach a significant level. After 21 days of drought, WL525 had the longest roots, which were significantly longer than those of the control group (*p* < 0.05).

All varieties also displayed larger root surface areas under drought stress than under normal watering conditions. After 14 days of drought, the root surface area of the seven varieties did not differ significantly from the control, although the overall trend showed an increase. Notably, after 21 days of drought, only the root surface area of WL525 was significantly higher than that of the control group (*p* < 0.05), while the root length of the other varieties was not significantly different from that of the control group. Changes in root volume were similarly observed. After 14 days of drought, the root volume of the Dimitra, Colosse, and WL525 varieties was significantly higher than that of the control group (*p* < 0.05), with Crown exhibiting the highest root volume. Following 21 days of drought, the root volume of Colosseo, Giant 801, PANGO, and Victoria fell below that of the control group. WL525 had the highest root volume, which was greater than the control, although the difference was not significant.

#### 2.2.2. Root Tip Number, Root Branch Number, and Root Diameter

Figure 3 presents the changes in root tip number, root diameter, and root branch number in the seven alfalfa varieties under drought stress. After 14 days of drought treatment, the root diameter of Crown was significantly higher than that of the control group (*p* < 0.05), but the root diameter of Giant 801 was significantly lower than that of the control group (*p* < 0.05), and the root diameter of crown was the largest. By 21 days of drought, the root diameter of Colosseo, Giant 801, Victoria, and WL525 was significantly lower than that of the control group, and the root diameter of WL525 was the largest.

Regarding root tip number, all varieties generally showed elevated values under drought compared with the control. After 14 days of stress, the root tip number of Dimitra and Crown was significantly higher than that of the control group (*p* < 0.05), and the root tip number of Crown was the largest. Following 21 days of drought, the root tip number of Colosseo and WL525 varieties was significantly higher than that of the control group, and the root tip number of Colosseo was the largest.

Drought also influenced root branch number. After 14 days of drought, the root branch number of Dimitra, Crown, and Giant 801 varieties was significantly higher than that of the control group (*p* < 0.05), and the root branch number of Crown was the largest. By 21 days of drought, the root branch number of PANGO varieties was significantly lower than that of the control group (*p* < 0.05), and the root branch number of WL525 was significantly higher than that of the control group, and the root branch number of WL525 was the largest. Overall, these results underscore substantial differences among varieties in root morphological adaptations, reflecting distinct drought resistance mechanisms.

### 2.3. Effects of Drought Stress on Physiological Indicators in Seven Alfalfa Varieties

#### 2.3.1. Antioxidant Enzymes

Figure 4 shows changes in antioxidant enzyme activities under drought stress, as well as the impact of extended stress duration. Drought generally stimulated higher peroxidase (POD), superoxide dismutase (SOD), and catalase (CAT) activities in all seven varieties, with peak values observed after 21 days of drought.

At 14 days of drought, the POD activity of Dimitra, Crown, Victoria, and WL525 was significantly higher than that of the control group (*p* < 0.05). Dimitra showed the highest POD activity. Following 21 days of drought, the POD activity of Crown, PANGO, and WL525 was significantly higher than that of the control group (*p* < 0.05), and PANGO had the highest POD activity.

At 14 days of drought, except for WL525 and Colosseo, the CAT activity of all varieties was significantly higher than that of the control group (*p* < 0.05), and the CAT activity of Crown was the highest. Following 21 days of drought, Dimitra died, the CAT activity of the other varieties was significantly higher than that of the control group (*p* < 0.05), and the CAT activity of WL525 was the highest.

At 14 days of drought, there was no significant difference in SOD activity between the treatment and control groups for all varieties (*p* > 0.05), and reached the highest in PANGO. Following 21 days of drought, the SOD activity of Colosseo, PANGO, and WL525 was significantly higher than that of the control group (*p* < 0.05), and the highest was obtained in PANGO.

#### 2.3.2. Effects of Drought Stress on H_2_O_2_, MDA, and Water Content in Seven Alfalfa Varieties

As shown in Figure 5, both hydrogen peroxide (H_2_O_2_) and malondialdehyde (MDA) contents were generally higher under drought stress compared with the control.

At 14 days of drought, the activity of H_2_O_2_ in Dimitra, Colosseo, and Victoria were significantly higher than that in the control group (*p* < 0.05), and the activity of H_2_O_2_ in Victoria was the highest. Following 21 days of drought, the H_2_O_2_ activity of Victoria and WL525 was significantly higher than that of the control group (*p* < 0.05), and Victoria had the highest H_2_O_2_ activity.

At 14 days of drought, the MDA content of Dimitra and Colosseo were significantly higher than that of the control group (*p* < 0.05), and the MDA content of Dimitra was the highest. Following 21 days of drought, the MDA content of Colosseo, Giant 801, and Victoria were significantly higher than that of the control group (*p* < 0.05), and the MDA content of Colosseo was the highest.

Water content (WC) did not differ significantly among the seven varieties after 14 days of drought. However, by 21 days, Colosseo showed the lowest WC. Overall, these results suggest that both antioxidant capacity and membrane stability markers vary significantly among alfalfa varieties in response to prolonged drought.

### 2.4. Correlation Analysis of Various Indicators in Seven Alfalfa Varieties Under Drought Stress

A correlation analysis (Figure 6) was conducted on 14 parameters, encompassing morphological and physiological indicators (plant height, aboveground biomass, root length, root surface area, root volume, root diameter, root tip number, root branch number, POD activity, CAT activity, SOD activity, H_2_O_2_ content, MDA content, WC). Aboveground biomass correlated significantly positively with root branch number, root tip number, root volume, plant height, and CAT (*p* < 0.01), but significantly negatively with MDA (*p* < 0.05). Root branch number was significantly positively correlated with root volume, aboveground biomass, plant height, root tip number, and CAT. Root diameter was significantly positively correlated with root surface area and WC, and negatively correlated with SOD. WC was significantly positively correlated with root surface area and root diameter. Root tip number showed significant positive correlations with root volume, root branch number, CAT, and plant height. Plant height correlated significantly positively with CAT, but significantly negatively with SOD and POD. MDA was significantly negatively correlated with root diameter, root surface area, root volume, WC, plant height, and aboveground biomass.

### 2.5. Principal Component Analysis

A principal component analysis (PCA) was performed on the 14 indicators (plant height, aboveground biomass, root length, root surface area, root volume, root diameter, root tip number, root branch number, POD activity, CAT activity, SOD activity, H_2_O_2_ content, MDA content, WC), yielding four principal components that collectively explained 90.436% of the total variance (Table 2). Principal Component 1 (47.97%), Principal Component 2 (21.114%), Principal Component 3 (10.977%), and Principal Component 4 (10.375%) had eigenvalues of 6.716, 2.956, 1.537, and 1.453, respectively, each exceeding the threshold eigenvalue of 1.

### 2.6. Membership Function Analysis of Drought Tolerance in Different Alfalfa Varieties

Based on the PCA results, the 14 indicators were grouped into four comprehensive factors. Their respective weights, calculated using Equation (2), were 0.4992 (Principal Component 1), 0.2197 (Principal Component 2), 0.1142 (Principal Component 3), and 0.1080 (Principal Component 4). Using the membership function method, the comprehensive drought tolerance (D) values for the seven alfalfa varieties were computed according to Equations (3) and (4).

As shown in Table 3, Crown had the highest D value (0.7013), indicating superior overall drought tolerance compared to the other varieties. Thus, Crown appears to possess the strongest adaptive mechanisms for withstanding drought among the tested alfalfa lines.

## 3. Discussion

### 3.1. Changes in Growth Parameters of Different Alfalfa Varieties Under Drought Stress

Drought stress is a critical environmental factor that severely affects plant germination, seedling development, growth, and overall survival, ultimately reducing biomass and productivity [23]. It has been extensively reported that drought disrupts assimilate allocation within plants [24]. To counteract such effects, plants often redistribute nutrients and biomass from leaves and stems to roots, thereby promoting increased root length and root surface area, which enhances overall plant vitality [16,25]. Our findings align with these general trends. All seven alfalfa varieties tested showed a reduction in plant height and aboveground biomass after 21 days of drought stress. However, a significant increase in root length, root surface area, and root tip number was observed compared to the control. This suggests that the plants responded to water deficiency by prioritizing root development to enhance water acquisition. It has been reported that a larger root size and a greater root dry weight can serve as selection criteria for identifying drought-tolerant plants [26,27]. Similar findings have also been reported in *Zea mays* L. Under water stress, *Zea mays* L. was found to have a greater rooting depth and was able to obtain more water from deep soil layers [28]. Comparable results were also observed in *Medicago truncatula*. It was found that under water stress, the TN6.18J genotype of *Medicago truncatula* had higher root biomass and greater root density. This indicates an effective strategy for adapting to scarce precipitation in the semi-arid environment of the Mediterranean [17]. However, additional experiments are needed to determine whether root activity in the tested varieties is similarly upregulated under drought conditions.

Plant height and biomass reductions under drought conditions are well-documented responses in many crops and forage species. For example, water stress often suppresses cell expansion and division, leading to reduced plant height and shoot biomass in *Zea mays* L. [29] and *Medicago truncatula* [29,30]. Studies have also reported that severe drought causes alterations in photosynthetic efficiency and hormonal signaling, further constraining shoot growth [31,32]. Similarly, biomass allocation shifts favoring root growth over shoot development have been observed in various leguminous forages, reflecting an adaptive mechanism to optimize water acquisition [33,34]. Marked differences in drought adaptation emerged among the seven alfalfa varieties. Compared with the other varieties, Crown and WL525 had higher values for plant height, aboveground biomass, and root morphological parameters. Their capacity to maintain these growth parameters under drought conditions underscores their stronger drought resilience. Colosseo and Victoria maintained relatively high plant height and aboveground biomass, but showed less pronounced changes in root morphology, suggesting moderate drought tolerance. In contrast, Giant 801 and PANGO exhibited smaller reductions in plant height and aboveground biomass, as well as only modest increases in root length and root tip number, reflecting higher sensitivity to drought. Dimitra did not survive beyond 21 days of drought stress, making it the most drought-sensitive variety. These findings are consistent with the comprehensive drought-tolerance rankings derived through membership function analysis, reinforcing the robustness of the evaluation method.

### 3.2. Changes in Physiological Parameters of Different Alfalfa Varieties Under Drought Stress

Plant drought tolerance closely correlates with antioxidant responses, as drought-resistant plants generally exhibit stronger antioxidant capacities that help sustain normal physiological processes [29,31]. The principal antioxidant enzymes include superoxide dismutase (SOD), catalase (CAT), and peroxidase (POD) [33]. It was reported that under drought stress, the activities of SOD, POD, and CAT in tobacco increased significantly, and the formation and removal of reactive oxygen free radicals were regulated by enhancing the activity of antioxidant enzymes, thus achieving a dynamic balance and enabling the plants to maintain their normal growth [35]. Accordingly, higher enzyme activities often signal better free-radical scavenging and, therefore, improved drought resistance. In this study, all seven varieties exhibited an increase in the activities of SOD, CAT, and POD under drought stress, indicating that all varieties possess the characteristic of adapting to drought by enhancing the activity of protective enzymes.

Lipid peroxidation is a hallmark of free radical-induced membrane damage under abiotic stress. Malondialdehyde (MDA) and hydrogen peroxide (H_2_O_2_) are major byproducts of membrane lipid peroxidation. MDA reflects the degree of lipid peroxidation. Its response under drought stress partially diverged from that of H_2_O_2_. Although H_2_O_2_ naturally arises from normal plant metabolism, excessive accumulation can be harmful [36]. It has been reported that under drought stress, the content of H_2_O_2_ and MDA in *S. davidii* and *Paspalum wettsteinii* increased significantly, and the activity of antioxidant enzymes (SOD, POD, CAT) also increased significantly [37]. Lipid peroxidation products will increase under drought stimulation, but plants eliminate free radicals by increasing antioxidant enzyme activity, maintaining internal balance, and ensuring the normal growth of plants [38]. In this study, H_2_O_2_ increased over the course of drought, peaking after 21 days. Crown, Giant 801, and WL525 demonstrated relatively lower H_2_O_2_ levels. After 14 days of drought, only Dimitra and Colosseo showed significantly elevated MDA, whereas the other varieties experienced more modest increases. Remarkably, Giant 801 exhibited a lower MDA concentration than the control, probably owing to heightened antioxidant enzyme activities at 14 days of drought likely due to their enhanced antioxidant enzyme activities that prevented severe membrane damage.

Water content (WC) is highly sensitive to drought stress, and is thus a critical factor in evaluating plant drought tolerance [39]. In the present experiment, WC in all seven varieties showed only a minor overall decrease, consistent with previous studies reporting water loss in alfalfa leaves under drought [40,41]. Under heat and drought co-stress, the relative water content of the barley-tolerant variety Lambada showed only a very weak downward trend compared to the control group. However, the relative water content of the sensitive variety, Spinner, showed a significant decrease compared to the control group [42], which strongly confirms the indicative role of moisture in reflecting plant resistance levels.

### 3.3. Comprehensive Evaluation of Drought Tolerance in Different Alfalfa Varieties

Drought tolerance is a complex, polygenic trait influenced by numerous morphological and physiological factors [43]. It has been reported that the evaluation of drought resistance of alfalfa is significantly correlated with many indexes, such as drought resistance index, greenhouse leaf senescence, leaf relative water content, canopy temperature, osmotic adjustment, leaf chlorophyll, protein, Ca, P, K, Mg, lignin, and so on. Therefore, using a single index to evaluate the drought resistance of alfalfa leads to significant bias [44]. A correlation analysis of this study revealed significant interdependencies among the 14 measured indicators, collectively shaping each variety’s drought response. Consequently, relying on any single indicator risks misrepresenting a variety’s true drought tolerance, emphasizing the importance of a multi-indicator approach to robustly evaluate drought resistance.

The combined application of principal component analysis (PCA) and membership function analysis has been widely adopted in assessments of drought tolerance. It was reported that in *Gleditsia sinensis*, seven physiological indexes were reduced to four principal components, and their drought resistance was comprehensively evaluated by membership function analysis [45]. In flue-cured tobacco, 21 individual indexes were reduced to 6 principal components, and their drought resistance was evaluated by membership function analysis [46]. Here, PCA distilled the interrelated variables into four principal components, each weighted according to its contribution. These weights informed the calculation of a comprehensive drought tolerance index (D value). Based on D values, the varieties ranked from most to least drought-resistant were as follows: Crown > WL525 > Colosseo > Victoria > PANGO > Giant 801 > Dimitra. The seven varieties were categorized into four groups: highly drought-tolerant (Crown and WL525), moderately drought-tolerant (Colosseo and Victoria), sensitive (PANGO and Giant 801), and highly sensitive (Dimitra). The comprehensive evaluation results were consistent with the preliminary drought tolerance rankings based on plant height, aboveground biomass, and root morphology, thereby validating the reliability of the comprehensive evaluation method.

## 4. Materials and Methods

### 4.1. Experimental Varieties and Seedling Cultivation

Seven alfalfa varieties—Dimitra, Crown, Colosseo, Giant 801, PANGO, Victoria, and WL525—were used as experimental materials. All seven varieties were provided by Guizhou Zhongzhiheng Ecology Co., Ltd. (Guiyang, China) and were successfully introduced and cultivated in Guizhou Province, demonstrating different production performance. Seeds were selected for plumpness and soaked in 0.1% sodium hypochlorite solution for 5 min. They were then rinsed 5 times with clean water until no residual sodium hypochlorite remained. The seeds were subsequently sown in plastic pots (45 cm × 25 cm × 20 cm), with 12 pots per variety and 20 plants per pot, resulting in a total of 240 plants per variety. The growth medium consisted of a 1:1 mixture of clay and nutrient soil. The clay was sourced from the College of Animal Science at Guizhou University, and the nutrient soil was obtained from the Guizhou Academy of Agricultural Sciences. Initially, the soil was fully irrigated with common irrigation water, reaching field capacity. Soil water content (SWC) was measured using a moisture meter (Wenzhou Weidu Electronic Co., Ltd., Wenzhou, China) in conjunction with the oven-drying method. The plants were watered completely every 5 days. They were grown in a greenhouse with a controlled temperature of 25 °C/20 °C, a 14 h light/10 h dark cycle, and 60% humidity. After six weeks of cultivation, the plant had grown 6–7 trifoliate leaves, marking the commencement of the experiment.

### 4.2. Experimental Design

In each variety, 180 seedlings, cultivated as described in Section 4.1, were randomly selected for transplanting. The transplanting conditions followed the same procedure as in Section 4.1, with 12 pots per variety and 15 plants per pot. A two-factor experimental design (time × treatment) was applied. The treatments included CK (control, watered completely once every 5 days) and drought (natural drought after complete watering). The time factors were 14 days and 21 days. Each treatment had three replicates, with one replicate per pot. Alfalfa plants were collected between 9:00 and 11:00 on the 14th and 21st days for both the control and drought stress treatments, for subsequent measurements.

### 4.3. Measurement Indicators and Methods

#### 4.3.1. Determination of Morphological Indicators

Ten plants from each variety and treatment were randomly selected for measurement. Plant height was measured to a precision of 0.1 cm using a ruler. Aboveground biomass was determined using an electronic balance after the aboveground portion of the plant had been oven-dried to a constant weight at 80 °C. Root length, root surface area, root volume, root branch number, root tip number, and root diameter were measured with an Epson Perfection V800 photo scanner (Epson, Suwa, Japan) in conjunction with Win-FOLIA Pro 2015 software (Regent Instruments, Québec City, QC, Canada).

Ten plants from each variety and treatment were randomly selected for measurement. Water content (WC) was calculated from fresh and dry weight measurements. Immediately after sampling, fresh weight (wf) was measured. Samples were then blanched in an oven at 105 °C for 15 min and dried to a constant weight at 80 °C (wd). Water content was calculated using Equation (1):WC = (wf − wd)/wf(1)
where wf is the fresh weight and wd is the dry weight.

#### 4.3.2. Determination of Physiological Indicators

For each treatment, three plants were randomly selected for measurement. The malondialdehyde (MDA) content was determined using the thiobarbituric acid (TBA) method. Fresh leaves (0.5 g) were ground in 10 mL of trichloroacetic acid (TCA) using a mortar and pestle. Then, 4 mL of 0.5% TBA was added to 1 mL of the supernatant. The mixture was heated, cooled, and centrifuged at 10,000× *g* for 5 min. The absorbance was measured at 532 nm [47]. To determine the hydrogen peroxide (H_2_O_2_) content, fresh samples (0.2 g) were extracted with 5 mL of 0.1% TCA (*w*/*v*), placed in an ice bath, and then centrifuged at 12,000× *g* for 15 min at 4 °C. Next, 0.5 mL of 100 mM phosphate buffer (pH 7.0) and 1 mL of 1 M potassium iodide were added to 0.5 mL of the supernatant. The absorbance was measured at 390 nm, and a standard curve was constructed to calculate the H_2_O_2_ content [48].

An amount of 0.3 g of plant samples was weighed and 5 mL of pre-cooled 50 mM sodium phosphate buffer were added (pH 7.8), containing 1 mM EDTA·Na^2^ and 2% PVP. The mixture was ground into a homogenate in an ice bath, then centrifuged at 12,000 rpm for 15 min at 4 °C. The supernatant obtained is the crude enzyme solution used for determining superoxide dismutase (SOD), peroxidase (POD), and catalase (CAT) activities.

Determination of SOD activity: A 3 mL reaction mixture containing 1.5 mL of 50 mM sodium phosphate buffer (pH 7.8), 0.25 mL distilled water, 0.3 mL of 130 mM methionine (Met), 0.3 mL of 750 µM nitrotetrazolium chloride (NBT), 0.3 mL of 100 µM EDTA·Na^2^, 0.3 mL of 20 µM riboflavin, and 0.05 mL of enzyme solution was prepared. Three test tubes were prepared: two with the reaction mixture and enzyme solution, and one with a control with sodium phosphate buffer instead of enzyme solution. After mixing, one control tube was placed in the dark and the other two tubes were placed at 25 °C under 300 μmol m^−2^ s^−1^ light for 5 min. The absorbance of the two test tubes was measured at 560 nm, using the dark tube as the blank. One unit of enzyme activity is defined as the 50% inhibition of photochemical reaction under the given conditions [49].

Determination of POD activity: A reaction mixture was prepared by adding guaiacol and 30% H_2_O_2_ to a 50 mM sodium phosphate buffer (pH 7.8) and dissolving the mixture. An amount of 3 mL of the reaction mixture and 0.1 mL sodium phosphate buffer were added to the control tube, and 3 mL of the reaction mixture and 0.1 mL enzyme solution were added to the sample tube. Immediately measure the absorbance at 470 nm, recording values every minute for a total of 5 min. The change in absorbance per minute of 0.01 is taken as one unit of enzyme activity [50].

Determination of CAT activity: A 3 mL reaction mixture was prepared, containing 1.5 mL of 50 mM sodium phosphate buffer (pH 7.8), 1 mL distilled water, 0.2 mL enzyme solution, and 0.3 mL of 0.1 mM H_2_O_2_. Distilled water (0.3 mL) was used in place of H_2_O_2_ for the control tube. The reaction was initiated by adding H_2_O_2_ to the sample tube, and then we immediately began timing. The absorbance was measured at 240 nm every minute for 3 min. One unit of enzyme activity is defined as the amount of enzyme that causes a 0.1 decrease in absorbance per minute [51].

### 4.4. Principal Component Analysis and Membership Function Analysis

The comprehensive evaluation of drought resistance in different alfalfa varieties was carried out according to the method described in [52], as follows:

Transforming multiple interrelated individual drought resistance indicators into independent comprehensive drought resistance indices X_j_ through principal component analysis.

Calculation of the weight (W_j_) for each comprehensive index using:(2)$$W_{j}=Vj÷\sum_{j=1}^{m}Vj$$
where W_j_ represents the weight of the jth comprehensive indicator among all indicators, and V_j_ represents the contribution rate of the jth comprehensive indicator.

Use of the membership function method to comprehensively analyze the drought resistance of alfalfa varieties to account for genetic differences. The membership function value R(X_j_) of each comprehensive index (X_j_) was calculated using the formula:R(X_j_) = (X_j_ − X_min_)/(X_max_ − X_min_)(3)
where R(X_j_) is the membership function value of the j comprehensive index, X_j_ is the value of the j comprehensive index, and X_min_ and X_max_ are the minimum and maximum values of the j comprehensive index among all tested materials.

Calculation of the comprehensive evaluation of drought resistance for different alfalfa varieties using the formula:(4)$$D=\sum_{j=1}^{n}\left[R(x_{j})\times W_{j}\right],j=1,2,\cdots ,n$$
where D represents the comprehensive evaluation value of drought resistance.

### 4.5. Data Analysis

Data were collated using WPS Office software and analyzed with SPSS 26.0 for two-factor statistical analysis. The least significant difference (LSD) method was used for multiple comparisons, with a significance level defined as *p* < 0.05. All data are presented as mean ± standard error, and graphics were generated using Origin 19.0 (OriginLab, Northampton, MA, USA).

## 5. Conclusions

This study investigated the growth and physiological responses of seven alfalfa varieties under drought stress by measuring eight morphological and six physiological indicators at both 14-day and 21-day drought intervals. Marked differences in drought tolerance emerged among the tested varieties. Principal component analysis and membership function analysis classified the seven varieties into four groups according to their drought tolerance: highly drought-tolerant (Crown and WL525), moderately drought-tolerant (Colosseo and Victoria), sensitive (PANGO and Giant 801), and highly sensitive (Dimitra). These alfalfa varieties adapt to drought by increasing root length, root surface area, and root tip number, enhancing protective enzyme activity, and reducing levels of hydrogen peroxide (H_2_O_2_) and malondialdehyde (MDA), thereby maintaining relatively high water content. These findings offer valuable insights into the drought-adaptive mechanisms of different alfalfa varieties and can guide breeding and cultivation decisions across diverse environmental settings. Notably, cultivating drought-tolerant varieties such as Crown and WL525 could significantly improve agricultural productivity and sustainability in regions prone to water scarcity. These results offer a scientific basis for the selection and breeding of alfalfa in karst landscapes and arid environments.

## Figures and Tables

**Figure 1 plants-14-00639-f001:**
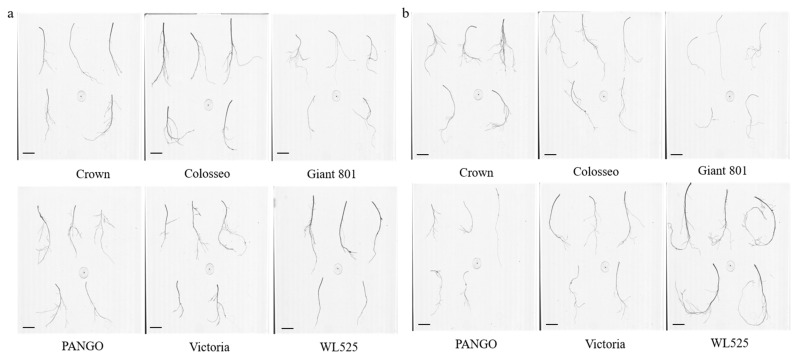
Root phenotype of seven alfalfa varieties under 21 days of (**a**) normal watering and (**b**) drought treatments. The bar: 5 cm.

**Figure 2 plants-14-00639-f002:**
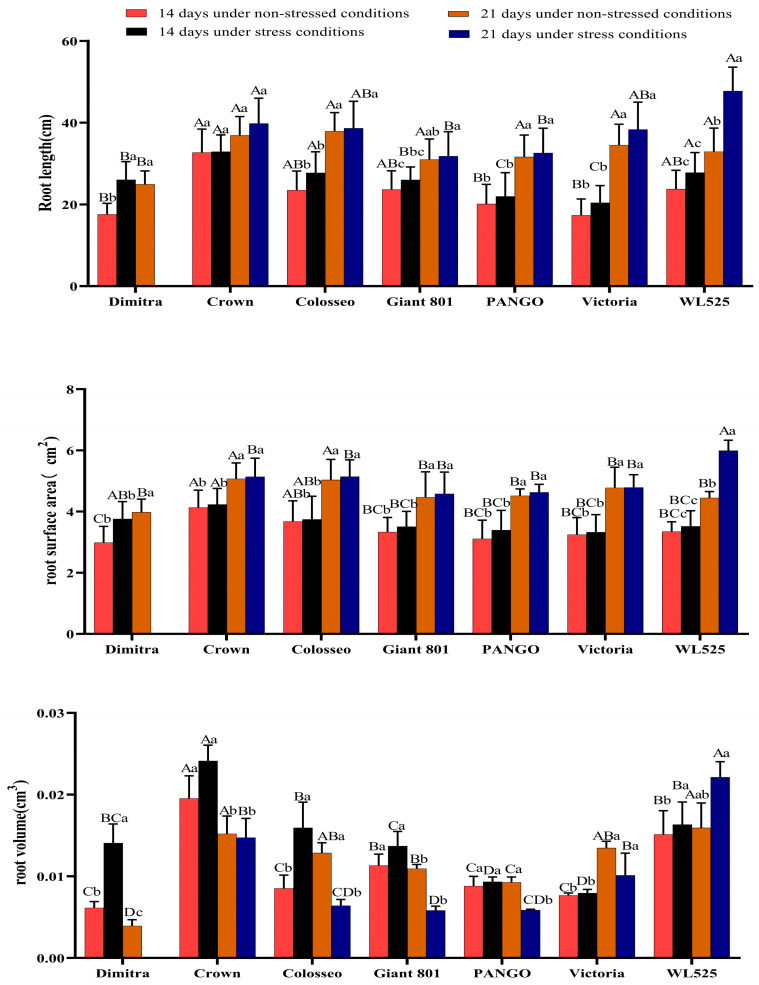
Effects of drought stress on root length, root surface area, and root volume in seven alfalfa varieties. Different uppercase letters indicate significant differences among varieties (*p* < 0.05). Different lowercase letters indicate significant differences between treatments (*p* < 0.05).

**Figure 3 plants-14-00639-f003:**
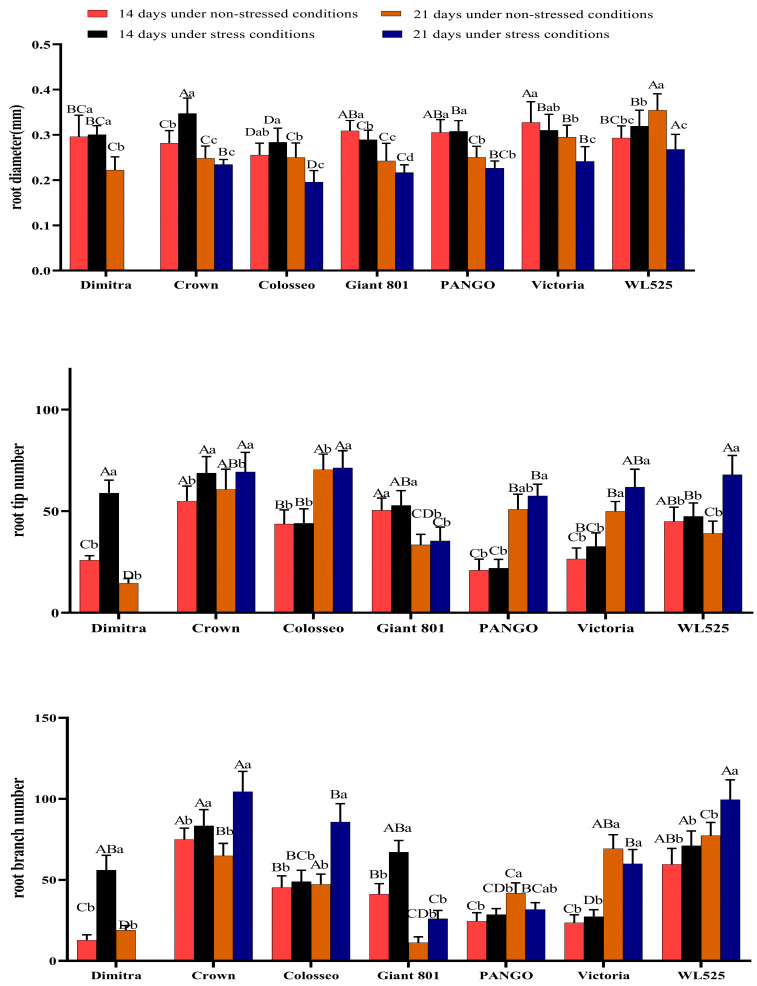
Effects of drought stress on root tip number, root branch number, and root diameter in seven alfalfa varieties. Different uppercase letters indicate significant differences among varieties (*p* < 0.05). Different lowercase letters indicate significant differences between treatments (*p* < 0.05).

**Figure 4 plants-14-00639-f004:**
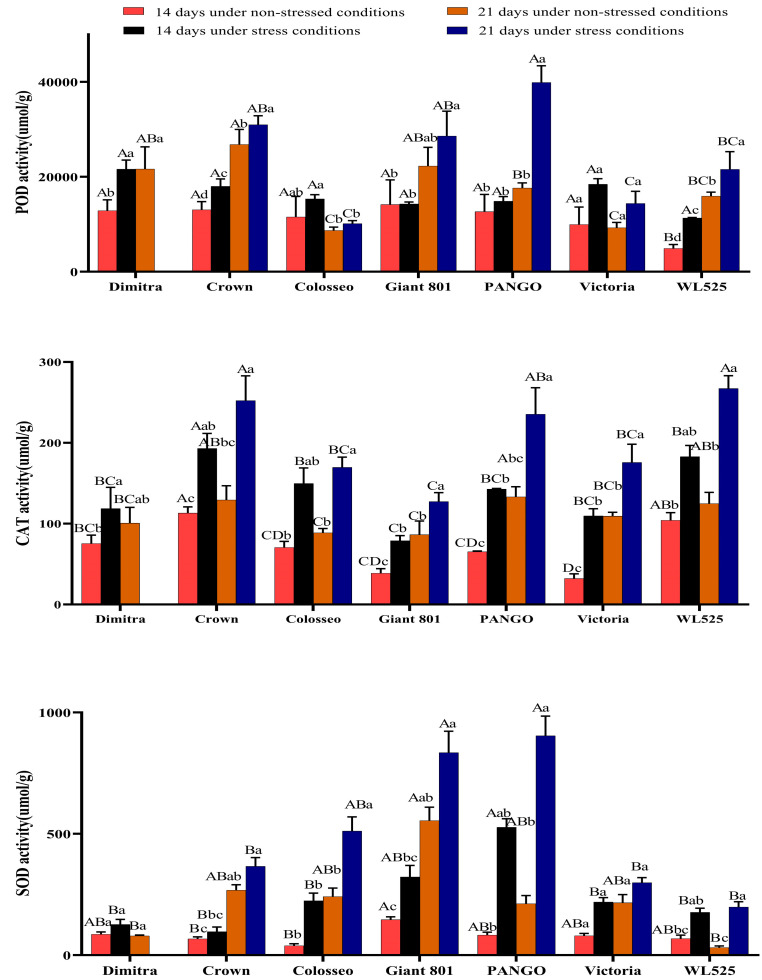
Effects of drought stress on antioxidant enzyme activities in seven alfalfa varieties. Different uppercase letters indicate significant differences among varieties (*p* < 0.05). Different lowercase letters indicate significant differences between treatments (*p* < 0.05).

**Figure 5 plants-14-00639-f005:**
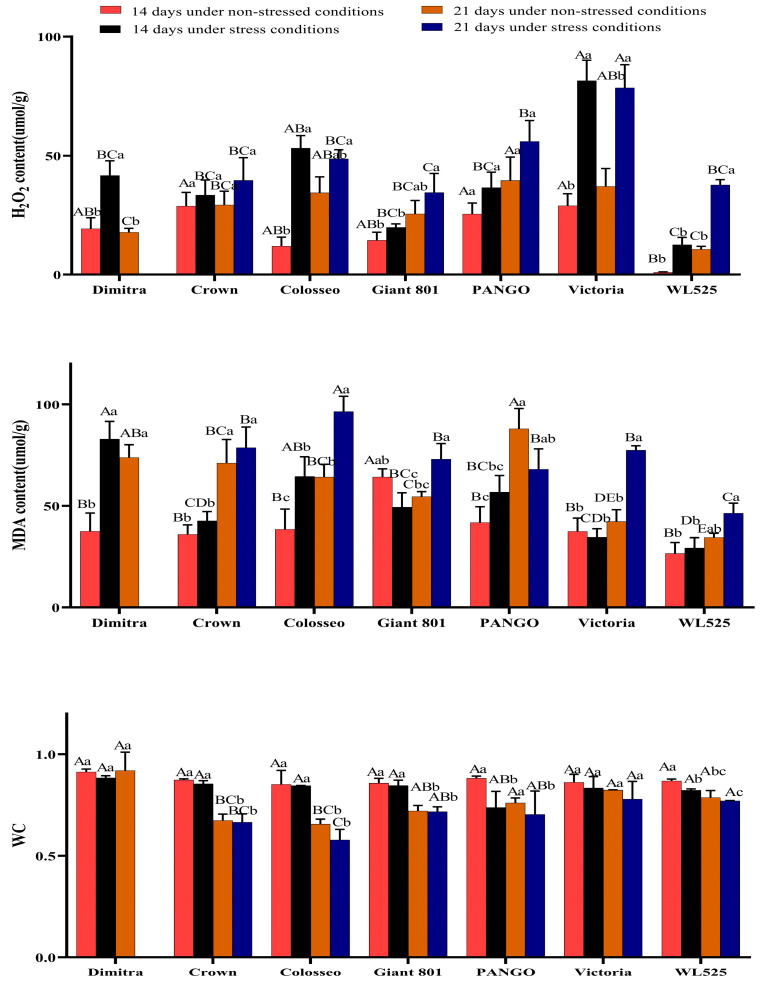
Effects of drought stress on H_2_O_2_, MDA, and WC in seven alfalfa varieties. Different uppercase letters indicate significant differences among varieties (*p* < 0.05). Different lowercase letters indicate significant differences between treatments (*p* < 0.05).

**Figure 6 plants-14-00639-f006:**
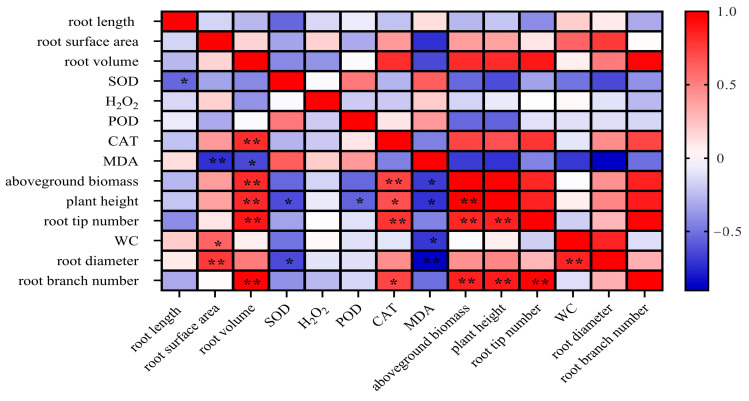
Correlation analysis of different indicators. * means significant correlation (*p* < 0.05), ** means extremely significant correlation (*p* < 0.01).

**Table 1 plants-14-00639-t001:** Effects of drought stress on plant height and aboveground biomass of seven alfalfa varieties.

Index	Variety	14 Days of Non-Stress Conditions	14 Days of Stress Conditions	Percentage Difference (%)	21 Days of Non-Stress Conditions	21 Days of Stress Conditions	Percentage Difference (%)
Plant height (cm)	Dimitra	10.74 ± 1.85 Ba	8.23 ± 2.02 Bb	76.6	11.97 ± 1.69 Ca	0	0.0
	Crown	20.96 ± 4.22 Ab	12.76 ± 1.88 Bc	60.9	25.38 ± 3.85 Aa	14.58 ± 4.02 Cc	57.4
	Colosseo	18.27 ± 3.21 Aa	14.2 ± 3.58 Ab	77.7	21.57 ± 3.79 Ba	13.69 ± 3.55 Cb	63.5
	Giant 801	10.8 ± 2.21 Ba	9.92 ± 2.18 Ba	91.9	11 ± 1.76 Ca	9.84 ± 2.67 Da	89.5
	PANGO	11.8 ± 2.39 Bab	9.85 ± 2.21 Bb	83.5	12.62 ± 3.52 Ca	7.69 ± 2.71 Dc	60.9
	Victoria	17.65 ± 3.73 Aa	10.21 ± 3.29 Bb	57.8	18.31 ± 3.12 Ba	17.61 ± 3.5 Ba	96.2
	WL525	19.7 ± 3.45 Ab	13.84 ± 3.55 Ac	70.3	26.19 ± 3.54 Aa	24.79 ± 3.68 Aa	94.7
Aboveground biomass (g)	Dimitra	0.026 ± 0.008 Ba	0.0294 ± 0.004 Aa	113.1	0.03 ± 0.006 Da	0	0.0
	Crown	0.143 ± 0.006 Aab	0.097 ± 0.003 Ab	67.8	0.216 ± 0.017 Ba	0.13 ± 0.08 Bab	60.2
	Colosseo	0.157 ± 0.009 Aab	0.093 ± 0.007 Ab	59.2	0.24 ± 0.015 Ba	0.124 ± 0.03 Bab	51.7
	Giant 801	0.039 ± 0.007 Ba	0.032 ± 0.006 Aa	82.1	0.042 ± 0.004 CDa	0.038 ± 0.017 Ca	90.5
	PANGO	0.039 ± 0.001 Ba	0.027 ± 0.006 Aa	69.2	0.18 ± 0.03 BCa	0.17 ± 0.027 Ba	94.4
	Victoria	0.065 ± 0.007 Ba	0.0225 ± 0.006 Ab	34.6	0.09 ± 0.006 CDa	0.0283 ± 0.001 Cb	31.4
	WL525	0.097 ± 0.003 ABb	0.061 ± 0.002 Ab	62.9	0.39 ± 0.015 Aa	0.27 ± 0.045 Aa	69.2

Different uppercase letters indicate significant differences among varieties (*p* < 0.05). Different lowercase letters indicate significant differences between treatments (*p* < 0.05). The percentage difference is the percentage of drought treatment values to control values. Values represent the mean ± standard error (SE).

**Table 2 plants-14-00639-t002:** Principal component analysis of the evaluated indicators.

Item	Principal Components			
	1	2	3	4
Eigenvalue	6.716	2.956	1.537	1.453
Contribution rate (%)	47.97	21.114	10.977	10.375
Cumulative contribution rate (%)	47.97	69.084	80.061	90.436

**Table 3 plants-14-00639-t003:** D values and comprehensive evaluation of different alfalfa varieties.

Variety	D	Overall Merit
Crown	0.7013	1
WL525	0.6987	2
Colosseo	0.4117	3
Victoria	0.3804	4
PANGO	0.3701	5
Giant 801	0.3132	6
Dimitra	0.1090	7

## Data Availability

The data are presented within the article.

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
