# Peer review of "Evaluation of Growth, Physiological, and Biochemical Responses of Different Medicago sativa L. Varieties Under Drought Stress"

_plants, 2025, doi:10.3390/plants14050639_

Round 1
Reviewer 1 Report
Comments and Suggestions for Authors
Evaluation of growth, physiological and biochemical responses of different Medicago sativa L. varieties under drought stress
Manuscript ID: plants-3422516
The manuscript, entitled Evaluation of growth, physiological and biochemical responses of different Medicago sativa L. varieties under drought stress. The manuscript presents the results of an experiment conducted in pots with alfalfa plants subjected to drought stress. In the context of growing global water deficits, climate change, it is imperative that this research topic is addressed. The article has correct structure, typical for scientific articles.
There are number of significant remarks that need to be taken into account.
Comment 1
The Introduction chapter is written in a too general a manner. The article provides minimal insight into the current state of knowledge regarding the impact of drought stress on alfalfa. It would be valuable to incorporate information regarding alfalfa's response to drought stress, including the compounds it produces under such conditions. Furthermore, he role of antioxidant enzymes such as SOD, POD, and CAT in tolerance to drought stress should be discussed.
Comment 2
P2 L 50 Contrary to the findings of the present study, Cuimei Zhang et al. (2019) reported that alfalfa exhibits good drought tolerance and can adapt to water-limited regions. Refer to this comment and revise the text
Comment 3
P2 L54-55 In my opinion, there is a substantial body of literature on the subject: Kang et. Al. 2011, He et al. 2012, Maghsoodi and Ramzmjoo 2015, Quan et al. 2016 etc. Refer to this comment and rephrase the text
Comment 4
P2 L 57-64 Elements of research with a more extensive global scope are absent. It is therefore difficult to ascertain the value of the article in terms of advancing the scientific field. It would be beneficial for the authors to highlight this in the text.
Comment 5
P2 L68 I would recommend writing about the reaction of varieties under normal watering.
Comment 6
P2 L 76 The authors present percentages in the text that are not reflected in the table. I suggest adding a column in Table 1 with the % results of plant height reduction relative to the control
Comment 7
P2 L 78-79 Write what was the reaction of Victoria and WL525
Comment 8
P3 L86 It is recommended that the "14 days of control" headings in Table 1 be combined into a single heading: 14 days of non-stress How the authors would explain the increase in Dimitra biomass after 14 days of stress relative to the control. This explanation should be incorporated into the text of the manuscript.
Comment 9
P3 L 92 Please insert a description of the results of the plants' reaction after 14 days of drought (changes in root length)
P3 L 94 ……..root length in all seven varieties ………
I suggest changing: ……root length in six varieties……..
Dimitra died
…….increased root length in all seven varieties compared with the control.
The observed differences were only significant for WL525; for the other varieties, the differences were not significant. Therefore, it is difficult to discuss and conclude that stress increased root length compared to the control. The same comment refers to P3 L 97-98
P3 L 97 Please insert a description of the results of the plants’ reaction after 14 days of drought (changes in surface area)
Comment 10
P3 L 90-101 The description of Figure 2 is insufficiently detailed. It is recommended that the description of the results be expanded.
Comment 11
P4 L109 …….most varieties exhibited thicker roots relative to the control, with Crown….
The differences were not statistically significant therefore it cannot be said that the results differed.
I suggest rewording the text
The same comment refers to P 4 L 111-112
Comment 12
P4 L 115-116 ….Dimitra increased its root tip number to 3.26 times that of the control.
What about the Crown variety, for it too, the difference was significant from the control
Comment 13
P 4 L 119-125 the same comment as comment 11
Comment 13
P7 L143 Please expand the description of the results to figure 4
Comment 14 P7 L143 At 14 days of drought Dimitra showed the highest POD activity…..
And what about Crown after 14 days of drought. The differences compared to controls are statistically significant
P7 L143 ……, Pango the highest SOD activity…..
And what about Giant 801?
Please refer to it in the text of manuscript.
Comment 14
P 8 L 17
Extend the description of the results to figure 5
Comment 15
P9 L176-178 In my opinion, root branch number exhibited a positive correlation with aboveground biomass.
Comment 16
P9 L185
Expand the description: ….indicators of morphology and physiology of Medicago sativa
Comment 17
P10 L230-232 ……..significant increases in root length, root surface area, and root tip number.
Compared to what? Add it in the text
Discussion
Comment 18
The discussion section needs to be modified. In the discussion section, the authors most often indicate that comparable findings to their results have been noted in other species for example P11 L273-274. Nevertheless, they do not undertake an analysis that would allow for the interpretation of their own results in relation to the world literature on the subject.
P11 L 283-284…..consistent with previous studies reporting water loss in alfalfa leaves under drought.
Cite the results of other researchers, provide numerical data where possible and try to discuss with your results
Materials and Methods
Comment 19
The manuscript contains numerous methodological flaws.
Why were these varieties chosen? What were their characteristics? What is their origin and stress tolerance? Add it in the text.
What was the experiment's design? Was randomization applied? P12 L 320-324
Comment 20
P 12 L 317 ……..watering was conducted every five days.
How much water was added to the pot, each time?
Where did plant growth take place? Greenhouse or growth chamber? How were the growth conditions of the plants maintained?
Comment 21
The number of plants in each repetition (P12 L313-314) and the conditions for plant growth during the experiment (temperature, lighting, photoperiod, and humidity %) were not specified in P12 L 315 as well as in P12 L 317. This is fundamental data for this type of experiment. Additionally, there is no information about the viability of the plants during the drought. The number of plants measured for the physiological and biochemical characteristics studied was not specified. The absence of these details renders the experiment difficult to reproduce. Add this information in the text.
Comment 22
The description of the determination of physiological indicators of plants is given minimal attention. In my opinion, this section of the text requires substantial supplementation P13 L 340-345
Describe the preparation of the samples
How much leaf tissue was taken?
What were the isolation and centrifugation conditions?
Comment 23
Describe in detail how SOD, POD, CAT, MDA and H2O2 were determined?
Comment 24
The authors conducted a statistical analysis of the data, but did not provide sufficient details on the what methods they used for statistical analyses. Add this information in the text.
Conclusions
Comment 25
Highlight the global significance of the results
Author Response
The manuscript presents the results of an experiment conducted in pots with alfalfa plants subjected to drought stress. In the context of growing global water deficits, climate change, it is imperative that this research topic is addressed. The article has correct structure, typical for scientific articles.
Comment 1: The Introduction chapter is written in a too general a manner. The article provides minimal insight into the current state of knowledge regarding the impact of drought stress on alfalfa. It would be valuable to incorporate information regarding alfalfa's response to drought stress, including the compounds it produces under such conditions. Furthermore, he role of antioxidant enzymes such as SOD, POD, and CAT in tolerance to drought stress should be discussed.
Response: Thank you for this valuable suggestion. In response, we have expanded the Introduction to provide a more comprehensive overview of the current understanding of alfalfa's drought response. We included information about the compounds produced by alfalfa under drought stress and highlighted the role of antioxidant enzymes (SOD, POD, and CAT) in drought tolerance. These revisions are marked in red font in the manuscript.
Comment 2: P2 L 50 Contrary to the findings of the present study, Cuimei Zhang et al. (2019) reported that alfalfa exhibits good drought tolerance and can adapt to water-limited regions. Refer to this comment and revise the text
Response: Thank you for your valuable suggestion. Following the suggestion and incorporating comment 1, we have re introduced the drought resistance of alfalfa in the original text and inserted new references to explain everything. The modified parts are marked in red font.
Comment 3: P2 L54-55 In my opinion, there is a substantial body of literature on the subject: Kang et. Al. 2011, He et al. 2012, Maghsoodi and Ramzmjoo 2015, Quan et al. 2016 etc. Refer to this comment and rephrase the text.
Response: We appreciate the reviewer’s input. We have revised the relevant sections of the text and incorporated the suggested references, ensuring that the discussion is more aligned with the broader body of literature. These changes are marked in red.
Comment 4: P2 L 57-64 Elements of research with a more extensive global scope are absent. It is therefore difficult to ascertain the value of the article in terms of advancing the scientific field. It would be beneficial for the authors to highlight this in the text.
Response: Thank you for the suggestion. We have added a discussion on the global relevance of our study and how it contributes to the broader understanding of drought stress in alfalfa. Relevant references have been included to underscore the global scope of the research. These changes are highlighted in red font.
Comment 5: P2 L68 I would recommend writing about the reaction of varieties under normal watering.
Response: We have followed your suggestion and expanded the text to include a description on the responses of the alfalfa varieties under normal watering conditions. The updated content is marked in red.
Comment 6: P2 L 76 The authors present percentages in the text that are not reflected in the table. I suggest adding a column in Table 1 with the % results of plant height reduction relative to the control.
Response: Thank you for this observation. We have added a column in Table 1 to show the percentage reduction in plant height relative to the control. The table has been updated, and the revisions are highlighted in red font.
Comment 7: P2 L 78-79 Write what was the reaction of Victoria and WL525
Response: Thank you for this suggestion. We have now included a detailed description of the response of both Victoria and WL525 varieties in the manuscript. These changes are marked in red font.
Comment 8: P3 L86 It is recommended that the "14 days of control" headings in Table 1 be combined into a single heading: 14 days of non-stress How the authors would explain the increase in Dimitra biomass after 14 days of stress relative to the control. This explanation should be incorporated into the text of the manuscript.
Response: For the consistency of the figures and tables throughout the text, we have added “Control is non-stress” in the remarks section of the tables and figures, and displayed them in red font. Regarding the Dimitra biomass increase, we have added an explanation in the text that although Dimitra biomass increased after 14 days of stress relative to the control, the difference did not reach statistical significance. This revision is marked in red font.
Comment 9: P3 L 92 Please insert a description of the results of the plants' reaction after 14 days of drought (changes in root length); P3 L 94 ……..root length in all seven varieties ………I suggest changing: ……root length in six varieties……..Dimitra died…….increased root length in all seven varieties compared with the control. The observed differences were only significant for WL525; for the other varieties, the differences were not significant. Therefore, it is difficult to discuss and conclude that stress increased root length compared to the control. The same comment refers to P3 L 97-98; P3 L 97 Please insert a description of the results of the plants’ reaction after 14 days of drought (changes in surface area).
Response: Thank you for your suggestions. We have clarified the description of the root length results, specifying that Dimitra died after 14 days of drought stress, and changed the wording to reflect the correct number of varieties (six, rather than seven). We removed the description of traits that did not show significant differences. And we have changed the way of description in the original text and marked it with red font. We have inserted a description of the 14-day root surface area and root volume in the original text and marked it with red fonts.
Comment 10: P3 L 90-101 The description of Figure 2 is insufficiently detailed. It is recommended that the description of the results be expanded.
Response: Thank you for the suggestion. In response, we have expanded the description of Figure 2 to provide a more detailed interpretation of the results. These changes are marked in red font.
Comment 11: P4 L109 …….most varieties exhibited thicker roots relative to the control, with Crown….,The differences were not statistically significant therefore it cannot be said that the results differed. I suggest rewording the text, The same comment refers to P 4 L 111-112.
Response: We have reworded the relevant sections of the manuscript to reflect that the differences in root thickness were not statistically significant, as per your suggestion. The revised text is marked in red font.
Comment 12: P4 L 115-116 ….Dimitra increased its root tip number to 3.26 times that of the control.What about the Crown variety, for it too, the difference was significant from the control
Response: Thank you for bringing this up. We have now included a description of the Crown variety’s response in the manuscript, noting that its root tip number also significantly increased compared to the control. This revision is marked in red font.
Comment 13: P7 L143 Please expand the description of the results to figure 4
Response: We have extended the description of Figure 4 to provide a more comprehensive interpretation of the results. These revisions are marked in red.
Comment 14: P7 L143 At 14 days of drought Dimitra showed the highest POD activity….. And what about Crown after 14 days of drought. The differences compared to controls are statistically significant. P7 L143 ……, Pango the highest SOD activity…...And what about Giant 801?Please refer to it in the text of manuscript.
Response: We have expanded the description in Figure 4 to include the statistical analysis for the Crown and Giant 801 varieties. We have also addressed the differences in POD and SOD activity among the varieties, as suggested. These updates are marked in red font.
Comment 15: P 8 L 17 Extend the description of the results to figure 5
Response: Thank you for your valuable suggestion. We have expanded the description of Figure 5 to provide more detailed context for the results. The revised text is marked in red font.
Comment 16: P9 L176-178 In my opinion, root branch number exhibited a positive correlation with aboveground biomass.
Response: We appreciate your input. Based on your suggestion, we have modified the manuscript to reflect the positive correlation between root branch number and aboveground biomass. The updated text is marked in red.
Comment 17: P9 L185, Expand the description: ….indicators of morphology and physiology of Medicago sativa
Response: Thank you for the suggestion. We have elaborated on the indicators of morphology and physiology in alfalfa, providing a more detailed explanation of their significance. These changes are marked in red font.
Comment 18: P10 L230-232 ……..significant increases in root length, root surface area, and root tip number. Compared to what? Add it in the text
Response: Thank you for this suggestion. We have clarified that the increases in root length, root surface area, and root tip number are compared to the control group. The revised text is marked in red font.
Comment 19: The discussion section needs to be modified. In the discussion section, the authors most often indicate that comparable findings to their results have been noted in other species for example P11 L273-274. Nevertheless, they do not undertake an analysis that would allow for the interpretation of their own results in relation to the world literature on the subject. P11 L 283-284…..consistent with previous studies reporting water loss in alfalfa leaves under drought.
Cite the results of other researchers, provide numerical data where possible and try to discuss with your results
Response: Thank you for this insightful comment. We have revised the discussion section to incorporate a more detailed analysis of our results in the context of global literature. We cited relevant studies and included numerical data where available to strengthen the comparison. These revisions are marked in red font.
Comment 20: The manuscript contains numerous methodological flaws.
Why were these varieties chosen? What were their characteristics? What is their origin and stress tolerance? Add it in the text.
What was the experiment's design? Was randomization applied? P12 L 320-324
Response: Thank you for your valuable suggestions. Dimitra, Crown, Colosseo, Giant 801, PANGO, Victoria, and WL525—were used as experimental materials, all of which have been successfully introduced and cultivated in Guizhou Province. Additionally, we have provided a description of the experimental design, including whether randomization was applied. The revised text is marked in red.
Comment 21: P 12 L 317 ……..watering was conducted every five days. How much water was added to the pot, each time? Where did plant growth take place? Greenhouse or growth chamber? How were the growth conditions of the plants maintained?
Response: Thank you for this clarification. The amount of water we irrigate each time reaches the field capacity. We have added the details about the growth environment, specifying the plants were grown in a greenhouse. We also clarified how the growth conditions (temperature, humidity, light, etc.) were controlled. These changes are marked in red font.
Comment 22: The number of plants in each repetition (P12 L313-314) and the conditions for plant growth during the experiment (temperature, lighting, photoperiod, and humidity %) were not specified in P12 L 315 as well as in P12 L 317. This is fundamental data for this type of experiment. Additionally, there is no information about the viability of the plants during the drought. The number of plants measured for the physiological and biochemical characteristics studied was not specified. The absence of these details renders the experiment difficult to reproduce. Add this information in the text.
Response: We appreciate the reviewer’s concerns. We have added the missing details regarding the number of plants per repetition, as well as information about the plant viability during the drought period. We also included specific growth conditions (temperature, lighting, photoperiod, humidity) and the number of plants used for physiological and biochemical measurements. The revised text is marked in red font.
Comment 23: The description of the determination of physiological indicators of plants is given minimal attention. In my opinion, this section of the text requires substantial supplementation P13 L 340-345.Describe the preparation of the samples. How much leaf tissue was taken? What were the isolation and centrifugation conditions?
Response: Thank you for your valuable suggestion. We have expanded the description of the physiological indicators, providing more details about sample preparation, the amount of leaf tissue used, and the isolation and centrifugation conditions. These revisions are marked in red font.
Comment 24: Describe in detail how SOD, POD, CAT, MDA and H2O2 were determined?
Response: Thank you for pointing this out. We have added detailed information about how SOD, POD, CAT, MDA, and H2O2 were measured, including the methods and procedures used for each determination. These changes are marked in red font.
Comment 25: The authors conducted a statistical analysis of the data, but did not provide sufficient details on the what methods they used for statistical analyses. Add this information in the text.
Response: We appreciate your comment. We have added a detailed description of the statistical methods used to analyze the data, including specific tests and statistical software. The revised text is marked in red font.
Comment 26: Highlight the global significance of the results
Response: Thank you for your suggestion. We have emphasized the global significance of our findings in the revised discussion section, particularly in relation to drought stress in alfalfa and its broader implications for agricultural sustainability. These revisions are marked in red font.

Reviewer 2 Report
Comments and Suggestions for Authors
Alfalfa is one of the most important forage crops in the world. Research into the mechanisms of alfalfa response to drought stress can contribute to the general understanding of how plants adapt to extreme environmental conditions. This in turn can lead to the development of new biotechnological and breeding strategies to increase crop resistance to drought.
The research results are very interesting and presented in an interesting way.
Comments:
Lines 75: in the Dimitra variety the decrease was smaller than in the PANGO variety
Lines 117: and Giant 801 and Colosseo
Lines 320: What phase of BBCH were the seedlings in?
In the discussion, the description of the research results is repeated. It should be reworded and more literature should be added to individual parameters, e.g. plant height and plant mass in drought conditions
Author Response
Comment 1: Lines 75: in the Dimitra variety the decrease was smaller than in the PANGO variety
Response: We appreciate your careful review of the data. Upon further verification, we discovered an error in our initial data calculations, which we have since corrected. The revised data now accurately reflects the observed differences between the Dimitra and PANGO varieties. We thank you again for bringing this to our attention.
Comment 2: Lines 117: and Giant 801 and Colosseo
Response: Thank you for your suggestion. In response to your comment, we have included a detailed analysis of the changes in root tip number for both Crown and Colosseo, as indicated by the updated data chart.
Comment 3: Lines 320: What phase of BBCH were the seedlings in?
Response: We greatly appreciate your insightful suggestion. The seedlings had reached the 6-7 trifoliate leaf stage after 6 weeks of cultivation. This information has now been explicitly added to the text and is highlighted in green for clarity.
Comment 4: In the discussion, the description of the research results is repeated. It should be reworded and more literature should be added to individual parameters, e.g. plant height and plant mass in drought conditions
Response: Thank you for pointing out the repetition in the discussion section. We have revised this part to streamline the description of our results and avoid redundancy. Additionally, we have integrated a more comprehensive review of the literature related to plant height and biomass under drought conditions, citing relevant studies to provide a broader context for our findings. These changes enhance the discussion and ensure a more thorough comparison with existing research.

Reviewer 3 Report
Comments and Suggestions for Authors
Review of the article Evaluation of Growth, Physiological and Biochemical Responses of Different Medicago Sativa L. varieties Under Drought Stress by Yang Wang et al.
Currently, research on drought resistance in agricultural plants is more relevant than ever. Climate change that is occurring around the world leads to frequent, prolonged droughts. Many regions suffer from a lack of water for irrigating crops. Such scientific research expands our understanding of how plants cope with stress.
This study is interesting, modern, and well-written. The reviewer believes that the article should be published. However, there are minor comments:
the methods do not say what kind of water was used to water the plants, distilled or regular irrigation water.
Author Response
Response to the reviewer 3:
Currently, research on drought resistance in agricultural plants is more relevant than ever. Climate change that is occurring around the world leads to frequent, prolonged droughts. Many regions suffer from a lack of water for irrigating crops. Such scientific research expands our understanding of how plants cope with stress.
Comment 1: the methods do not say what kind of water was used to water the plants, distilled or regular irrigation water.
Response: Thank you for your insightful comment. We have updated the methods section to clarify the type of water used for irrigation. Specifically, we used regular irrigation water throughout the experiment. This addition has been marked in blue font for easy reference.

Round 2
Reviewer 1 Report
Comments and Suggestions for Authors
Evaluation of growth, physiological and biochemical responses of different Medicago sativa L. varieties under drought stress
Manuscript ID: plants-3422516
The manuscript, entitled Evaluation of growth, physiological and biochemical responses of different Medicago sativa L. varieties under drought stress. The manuscript has undergone significant revisions from the original version; however, it still exhibits a number of deficiencies. It is still in need of improvement and completion.
Comment 1
P2 L 65-70 and L78-83 The authors have included the text on the broader global impact of the study; however, some statements are factually inaccurate and overly extensive in their scope.
For example: ………offering valuable insights for improving drought resilience in alfalfa cultivation…..
In my opinion your study does not improve resistance only indicates more tolerant varieties
For example ………..broader understanding of how alfalfa can be cultivated sustainably in regions ………..
In my opinion your study identifies varieties that are more tolerant to water stress. The manuscript does not include any recommendation regarding the cultivation of alfalfa
Comment 2
P2 L 74 ……….(2) elucidate their physiological mechanisms of drought tolerance……
In my opinion the study determines the level of physiological indicators in plants subjected to drought stress rather than explaining the physiological mechanisms
Comment 3
P3 L90-93 I suggest this sentence be moved higher after the sentence ending: increased over time.
Comment 4
P3 L 97 …….decline retaining only 57.8%.....
I suggest changing the retaining to reaching
Comment 5
P3 L94 After 14 days of drought stress, plant height significantly decreased in five of the……
Give reference, table ?
Comment 6
I suggest changing the headings in Table 1
“14 days of control” to “14 days under non-stressed conditions”
“14 days of drought” to “14 days under stress conditions”
Letter missing at 0.0225±0.006A… value in table 1
Comment 7
P4 L122 Delete “After 14 days of drought”
Comment 8
P4 L 123 …..length, but only the root length of Dimitra ………
Comment 9
P4 L 129 ……the surface area only of WL525 was ……
P5 L 144 and 146 Crown instead of crown
Comment 10
P6 Figure 2
I suggest changing the description of the bars: 14 days of non-stress conditions, 21 days of non-stress conditions
Comment refers to figures 2, 3, 4, 5
Comment 11
P11 L 239 Expand the description of the figure: ….indicators of morphology and physiology of Medicago sativa
Comment 12
P12 L 300
The cited literature refers to other species not alfalfa but [29] Zea Mays and [30] Medicago trunculata.
Correct it in the text.
Comment 13
P12 L298-300 , P12 L300-301, P12 L 302-304
Where possible, add numerical values to the text and interpret them in relation to your own research.
Comment 14
P13 L338-342
Where possible, add numerical values to the text and interpret them in relation to your own research.
Comment 15
P13 L343-348
Please interpret your own results in relation to the literature
Comment 16
P13 L349 -352
In my opinion, the statement that your results are in agreement with the result of other authors reporting water loss in alfalfa leaves under drought is too vague, insufficient.
Try to interpret your results in comparison with other results so that the reader can draw conclusions from the study
Comment 17
Where possible, add numerical values to the text and interpret them in relation to your own research.
Comment 18
P14 L365-377
Please interpret your own results in relation to the literature
Comment 19
P14 L380-382
Why were these varieties chosen? What were their characteristics? What is their origin and stress tolerance? Add it in the text.
Comment 20
P14 L 383
specify in the text 3 or 5 min
Comment 21
P17 L 495
It is worth mentioning what are the drought-adaptive mechanisms of alfalfa varieties
Author Response
Comment 1: P2 L 65-70 and L78-83 The authors have included the text on the broader global impact of the study; however, some statements are factually inaccurate and overly extensive in their scope.
For example: ………offering valuable insights for improving drought resilience in alfalfa cultivation….. In my opinion your study does not improve resistance only indicates more tolerant varieties
For example ………..broader understanding of how alfalfa can be cultivated sustainably in regions ………..In my opinion your study identifies varieties that are more tolerant to water stress. The manuscript does not include any recommendation regarding the cultivation of alfalfa
Response: Thank you for your valuable suggestion. We have revised the introduction according to your suggestion to accurately reflect our findings, emphasizing that this study identified more drought tolerant alfalfa varieties rather than improving drought resistance. These modifications are marked in red in the revised manuscript.
Comment 2: P2 L 74 ………. (2) elucidate their physiological mechanisms of drought tolerance……
In my opinion the study determines the level of physiological indicators in plants subjected to drought stress rather than explaining the physiological mechanisms
Response: Thank you for your insightful comment. We have revised this statement to more accurately describe the study’s scope. The revised text has been highlighted in red in the manuscript.
Comment 3: P3 L90-93 I suggest this sentence be moved higher after the sentence ending: increased over time.
Response: We appreciate the suggestion. The sentence has been repositioned as recommended and marked in red in the manuscript.
Comment 4: P3 L 97 …….decline retaining only 57.8%.....
I suggest changing the retaining to reaching
Response: Thank you for your valuable suggestions. Thank you for your valuable suggestions. It was modified in the original text. Mark with red font.
Comment 5: P3 L94 After 14 days of drought stress, plant height significantly decreased in five of the……
Give reference, table ?
Response: Thank you for your valuable suggestions. It was modified in the original text. Mark with red font.
Comment 6: I suggest changing the headings in Table 1
“14 days of control” to “14 days under non-stressed conditions”
“14 days of drought” to “14 days under stress conditions”
Letter missing at 0.0225±0.006A… value in table 1
Response: Thank you for your valuable suggestions. We have modified the headings in Table 1 according to your suggestion. We have corrected the missing letter in the table and marked the change in red.
Comment 7: P4 L122 Delete “After 14 days of drought”
Response: Thank you for your valuable suggestions. We have deleted ' After 14 days of drought ' in the original.
Comment 8: P4 L 123 …..length, but only the root length of Dimitra ………?
Response: Thank you for your valuable suggestions. MayIt was modified in the original text. Mark with red font.
Comment 9: P4 L 129 ……the surface area only of WL525 was ……
P5 L 144 and 146 Crown instead of crown
Response: Thank you for your valuable suggestions. It was modified in the original text. Mark with red font.
Comment 10: P6 Figure 2
I suggest changing the description of the bars: 14 days of non-stress conditions, 21 days of non-stress conditions
Comment refers to figures 2, 3, 4, 5
Response: Thank you for your valuable suggestions. We have made the modifications in figure 2-5 according to your suggestions.
Comment 11: P11 L 239 Expand the description of the figure: ….indicators of morphology and physiology of Medicago sativa
Response: Thank you for your valuable suggestions. In accordance with your comments, we have modified the description of the original text and marked it with red font.
Comment 12: P12 L 300
The cited literature refers to other species not alfalfa but [29] Zea Mays and [30] Medicago trunculata.
Correct it in the text.
Response: Thank you for your valuable suggestions. In accordance with your comments, we have modified the description of the original text and marked it with red font.
Comment 13: P12 L298-300, P12 L300-301, P12 L 302-304
Where possible, add numerical values to the text and interpret them in relation to your own research.
Response: Thank you for your valuable suggestions. There is no specific description of each indicator data in the references, only trend and difference descriptions, so it is difficult to find and cite specific indicator values.
Comment 14: P13 L338-342
Where possible, add numerical values to the text and interpret them in relation to your own research.
Response: Thank you for your valuable suggestion. We tried our best to search for data descriptions in the original references and further improved the description of the data in the citations. The modified parts are marked in red.
Comment 15: P13 L343-348
Please interpret your own results in relation to the literature
Response: Thank you for your valuable suggestions. We have modified the original text and marked it with red font.
Comment 16: P13 L349 -352
In my opinion, the statement that your results are in agreement with the result of other authors reporting water loss in alfalfa leaves under drought is too vague, insufficient.
Try to interpret your results in comparison with other results so that the reader can draw conclusions from the study
Response: Thank you for your valuable suggestions. We have modified the original text and marked it with red font.
Comment 17: Where possible, add numerical values to the text and interpret them in relation to your own research.
Response: Thank you for your valuable suggestions. We have modified the original text and marked it with red font.
Comment 18: P14 L365-377
Please interpret your own results in relation to the literature
Response: Thank you for your valuable suggestions. We have modified the original text and marked it with red font.
Comment 19: P14 L380-382
Why were these varieties chosen? What were their characteristics? What is their origin and stress tolerance? Add it in the text.
Response: Thank you for your valuable suggestions. We have added more details regarding the selection of the alfalfa varieties, including their characteristics and origin, as per your suggestion. These additions are marked in red.
Comment 20: P14 L 383
specify in the text 3 or 5 min
响应: 感谢您的宝贵建议。我们修改了原始文本并用红色字体标记。
评论 21: P17 L 495
值得一提的是紫花苜蓿品种的适应干旱机制有哪些
响应: 感谢您的宝贵建议。本研究对紫花苜蓿抗旱机制进行了总结和补充,以红色字体显示。

Reviewer 2 Report
Comments and Suggestions for Authors
The manuscript has been thoroughly revised. In the current version it is possible to publish.
Only on line 122 you need to remove: 'After 14 days of drought'
Author Response
评论 1:仅在第 122 行,您需要删除:“干旱 14 天后”
响应: 谢谢你的建议。我们已根据要求删除了“干旱 14 天后”一词。该更改在手稿中标记为红色。
